# Individualized Dosage Optimization for Myeloablative Conditioning before Unrelated Cord Blood Transplantation in a Diamond–Blackfan Anemia Patient with Germline *RPL11* Mutation: A Case Study

**Rong-Long Chen** [1,*] **, Li-Hua Fang** [2] **and Liuh-Yow Chen** [3,*]

1   Department of Pediatric Hematology and Oncology, Koo Foundation Sun Yat-Sen Cancer Center, Taipei City 112, Taiwan
2   Department of Pharmacy, Koo Foundation Sun Yat-Sen Cancer Center, Taipei City 112, Taiwan; lihua@kfsyscc.org
3   Institute of Molecular Biology, Academia Sinica, Taipei City 115, Taiwan
*   Correspondence: rlchen@kfsyscc.org (R.-L.C.); lyowchen@gate.sinica.edu.tw (L.-Y.C.);
    Tel.: +886-2-2897-0011 (R.-L.C. & L.-Y.C.)

**Abstract:** Unrelated cord blood transplantation (CBT) for Diamond–Blackfan anemia (DBA), a systemic ribosomopathy affecting the disposition of conditioning agents, has resulted in outcomes inferior to those by transplantations from matched donors. We report the experience of the pharmacokinetics-guided myeloablative unrelated CBT in a DBA patient with a germline *RPL11* mutation. The conditioning consisted of individualized dosing of fludarabine (based on weight and renal function with a target area under the curve (AUC) of 17.5 mg·h/L) and busulfan (based on therapeutic drug monitoring with a target AUC of 90 mg·h/L), as well as dosing and timing of thymoglobulin (based on body weight and pre-dose lymphocyte count to target pre-CBT AUC of 30.7 AU·day/mL and post-CBT AUC of 4.3 AU·day/mL, respectively). The pharmacokinetic measures resulted in a 27.5% reduction in busulfan and a 35% increase in fludarabine, as well as an over three-fold increase in thymoglobulin dosage with the start time changed to day-9 instead of day-2 compared to regular regimens. The transplantation resulted in rapid, complete, and sustained hematopoietic engraftment. The patient is now healthy over 3 years after CBT. A pharmacokinetics-guided individualized dosing strategy for conditioning might be a feasible option to improve the outcomes of DBA patients receiving unrelated myeloablative CBT.

**Keywords:** population pharmacokinetics model; therapeutic drug monitoring; Diamond–Blackfan anemia; unrelated cord blood transplantation; myeloablative conditioning

## 1. Introduction

Diamond–Blackfan anemia (DBA), a constitutional inherited bone marrow failure syndrome, is characterized by erythroblastopenia revealed by signs of early-onset anemia and various congenital malformations [1,2]. Germline variants resulting from different gene mutations in one of the ribosomal protein (RP) genes (numbered up to 20) or in a gene involved in ribosome biogenesis (namely *TSR2*) have been implicated in classical DBA. These result in the defect in ribosomal RNA maturation, generating nucleolar stress and leading to cell cycle arrest and apoptosis [3]. Approximately half of all DBA patients achieve an adequate response to glucocorticoids with transfusion independence [1,4]. Hematopoietic cell transplantation (HCT), mostly from related or unrelated human leukocyte antigen (HLA)-matched donors, is a curative option for the severe hematological changes seen in Diamond–Blackfan anemia with transfusion dependency or glucocorticoid toxicities [1]. However, the difference in pharmacokinetics (PK) of conditioning agents can be associated

with variable myeloablation and immune suppression before and after cord blood transplantation (CBT), resulting in different outcomes [5]. This may be particularly true in DBA patients with systemic ribosomopathy, as high rates of graft failure and other complications, such as sinusoidal obstruction syndrome and respiratory distress syndrome, happened after unrelated CBT [6,7]. We hypothesized that the systemic ribosomopathy, being reported to result in short stature, congenital anomalies, organ malformations as well as developmental abnormalities, may also affect the drug disposition and compromise the outcomes of unrelated CBT, which requires the strict dosage optimization of conditioning agents. Processes including individually optimized dosages of fludarabine/thymoglobulin based on the population PK models and of busulfan by therapeutic drug monitoring (TDM) in the myeloablative conditioning regimens before unrelated CBT may improve the outcomes. Herein, we report the case of a DBA patient with a germline *RPL11* mutation who was successfully treated by PK-guided individualized conditioning before myeloablative unrelated CBT at about six years of age when serious glucocorticoid side effects and progressive iron overload had evolved.

## 2. Materials and Methods

### 2.1. Whole Exome Sequencing (WES)

To identify genetic variants in the patient, genomic DNA from leukocytes of the subject was subjected to WES. The library for sequencing was generated using TruSeq DNA Exome kit (Illumina, San Diego, CA, USA). WES was performed using NextSeq 550 instrument (Illumina, San Diego, CA, USA). For bioinformatics analysis, low-quality bases and sequencing adapters in raw data were removed using Trimmomatic (v.0.36), and then the reads were aligned to a reference genome (hg19) using the Burrows–Wheeler aligner (BWA, v.0.7.17). The results of the alignment step were recorded in .bam format, and then the .bam file was processed using Picard tools (v.2.17.2), with processes like sorting and duplicate marking. After that, the variant calling step was performed by GATK (v.4.0.4.0) with HaplotypeCaller task (v.4.0.4.0), and variants were annotated by VEP (v.92). Analysis of the variants in the genes associated with DBA led to the identification of *RPL11* c.95_96del,p.(Arg32ThrfsTer22) mutation. Sanger sequencing, using an Applied Biosystems 3730xl DNA analyzer, was carried out to verify the *RPL11* mutation. A fragment of DNA containing the *RPL11* mutation site was amplified by PCR using oligonucleotide primers 5′-GATCAAGGTGAAAAGGAGAACC-3′ and 5′-CTTTTCATTTCTCCGGATGCC-3′, which were also used for direct sequencing. Our results confirm the *RPL11* mutation in the patient and revealed the mother had wild-type RPL11 alleles.

### 2.2. Fludarabine Dose Optimization

Fludarabine TDM was done according to the TARGET study protocol (EudraCT 2018-000356-18) but in a slightly modified manner to allow for better AUC calculations. InsightRX (San Francisco, CA, USA) was then used to target fludarabine cumulative AUC at around 20 mg·h/L (range 17.5–22.5 mg·h/L) using the PK model. Using this model, the individual's patient data (age, sex, weight, height, and serum creatinine) were incorporated to estimate the cumulative AUC using a maximum a posteriori probability (MAP) Bayesian fitting. The estimation was then checked for goodness of fit within InsightRX, both at the overall level as well as at the individual sample level [8], Thus, instead of the body surface area, the actual body weight (10.5 kg) and renal function (serum creatinine, 0.42 mg/dL) were used to estimate the measures of fludarabine exposure. The target AUC of 17.5 mg·h/L was chosen for having the best relationship with overall toxicity/efficacy and having a hypothesized capability of in vivo depletion of T cells [9]. Thus, the actual given dosage of fludarabine for the patient according to the model was 216 mg/m$^2$ divided into four daily doses.

### 2.3. TDM of Busulfan

The protocol has been described in detail in [10]. Briefly, 50 μL plasma samples (obtained from the patient collected for busulfan TDM at 5 min, 1 h, 2 h, and 3 h after busulfan infusion) were mixed with the following reagents in sequence: 12.5 μL ISTD working solution, 12.5 μL of 50% ACN, and 25 μL of 20% trichloroacetic acid (20%). The mixtures were vortexed for 60 s and centrifuged for 5 min at 10,000× *g*. The supernatant (60 μL) was transferred into a new glass vial containing 540 μL of 5% ACN. The samples were analyzed by performing LC–MS/MS 24 h after preparation using the TSQ Quantiva triple quadrupole mass spectrometer (Thermo Fisher Scientific, Waltham, MA, USA), which was coupled online with the UltiMate 3000 Open Sampler XRS System (Thermo Fisher Scientific). LC separation was performed as per the methods previously described [11]. Eight microliters of the sample were loaded into an ACQUITY UPLC BEH C18 Column (130 Å, 1.7 μm, 2.1 mm × 50 mm; Waters, Milford, MA, USA). Mobile phase A comprised 0.1% formic acid in water, and mobile phase B included acetonitrile with 0.1% ammonium acetate. The elution program was conducted for 0–2.4 min in isocratic 5% B, for 2.4–2.75 min from 5% to 95% B, for 2.75–3.5 min in isocratic 95% B, for 3.5–4.0 min from 95% to 5% B, and for 4.0–4.5 min in isocratic 5% B, with a flow rate 0.7 mL/min throughout the analyses. The ion source parameters were as follows: sheath gas of 38 AU, aux gas of 12 AU, ion transfer tube temperature of 250 °C, and vaporizer temperature of 279 °C. The busulfan and busulfan-D8 were analyzed in the selected reaction monitoring mode with the following transitions: busulfan 264 > 151 $m/z$ (quantifier) and 264 > 151 $m/z$ (qualifier) (CE: 10 V, RF: 50 V), as well as busulfan-D8 272 > 159 $m/z$ (quantifier) (CE: 11 V, RF: 48 V) and 272 > 62 $m/z$ (qualifier) (CE: 23 V, RF: 48 V). A dwell time of 50 ms was observed for both the Q1 and Q3 resolutions (FWHM) of 0.7 Da, CID gas of 1.5 mTorr, source fragmentation of 10 V, and chrom filter of 3 s. The dosing of intravenous busulfan, based on therapeutic drug monitoring, was determined using a validated busulfan population PK model, which is available in the cloud-based InsightRX Platform (San Francisco, CA, USA). The results from the busulfan TDM were fitted into the busulfan population PK model through the model-based Bayesian forecasting method, and a myeloablative cumulative AUC of 90 mg·h/L was obtained [10,12]. Thus, the actual given dosage of busulfan for this patient was 348 mg/m$^2$.

### 2.4. Thymoglobulin Dose Optimization

Based on a cohort of 260 patients with ages varying from 0.1–21 years old, a population PK model was developed to both describe the PK of thymoglobulin and the effect of growing up on the PK parameters [13–15]. The developed model could well predict thymoglobulin concentrations in all age groups and was extensively validated using resampling and simulation techniques. Two variables, the body weight of 10.5 kg (clearance and volume of distribution) and the pre-conditioning peripheral blood lymphocyte count of 4850 per μL (clearance), clearly influenced the PK. Using the developed PK model and the known therapeutic window, an individualized dose of thymoglobulin can be determined based on body weight and pre-thymoglobulin peripheral blood lymphocyte counts for the patient. Then, the simulation studies were performed, with the thymoglobulin dose and timing varied relative to the HCT, while fixing the patient- and treatment-specific details to this patient, aiming for an optimal pre- and post-HCT exposure. A total of 1000 simulations were performed, with full inter-individual variability, and the median and 75% confidence interval were derived. The thymoglobulin dosage (15 mg/kg divided into four daily doses starting from day-9) was chosen, targeting AUC of 30.7 AU·day/mL pre-CBT and of 4.3 AU·day/mL post-CBT, respectively.

## 3. Detailed Case Description

The patient and his family members gave their informed consent for inclusion before they participated in this study. The protocol was approved by the Ethics Committee of the Koo Foundation Sun Yat-Sen Cancer Center (IRB approval code: 20131209A). The male patient was born at 37 weeks of gestation with a birth weight of 3200 g. He was

diagnosed with heart failure soon after birth when cardiac sonography showed patent ductus arteriosus, ventricular septal defect, patent foramen ovale, and pulmonary stenosis. The initial hemoglobin concentration was 2.3 mg/dL with thrombocytopenia (platelets $46 \times 10^9$/L). His condition stabilized gradually after supportive treatment, including four 10–20 mL per kg of red blood cell transfusions. Follow-up cardiac sonography only revealed mild mitral regurgitation without other abnormalities.

DBA was diagnosed soon after because of characteristic reticulocytopenia macrocytic anemia, hypocellular marrow, and pure erythroid hypoplasia. Oral prednisolone was administered after diagnosis and further transfusions were avoided by prescribing a daily treatment of 1 to 2 mg/kg prednisolone for more than 5 years. During that time, the patient exhibited severe growth retardation, but cognition and developmental milestones were normal.

He suffered progressive back pain since 4 years of age and was informed that he had complications including prednisolone-related osteoporosis with L1-spine compression fracture and bilateral cataracts when he was 5 years of age. Prednisolone taper was initiated and red blood cell transfusions were re-started. However, he encountered an episode of adrenal crisis with severe fatigue, hypotension, hypoglycemia, and electrolyte imbalance. He was transferred to our facility at 5.5 years of age after receiving red blood cell transfusions four times (about 20 mL per kg each time).

Upon arrival, he had bilateral cataracts and extreme growth retardation (far below the third percentile of age-appropriate height and weight), as well as scoliosis with wedge deformities of the T12-L1 vertebrae. Tertiary adrenal insufficiency was noted, with low cortisol and undetectable adrenocorticotropic hormone. He required a cortisone maintenance treatment of 12.5 mg twice daily. Given this low steroid dose, he required red blood cell transfusions every 3 to 5 weeks to maintain the hemoglobin levels above 7 mg/dL. He exhibited iron overload upon arrival and elevated ferritin (1099 ng/mL). He presented no other anomalies or organ dysfunctions. We identified a previously reported *RPL11* c.95_96del,p.(Arg32ThrfsTer22) mutation [16] through whole-exome sequencing, followed by Sanger sequencing validation (Figure 1). Both parents were phenotypically normal, and his mother did not carry the mutation.

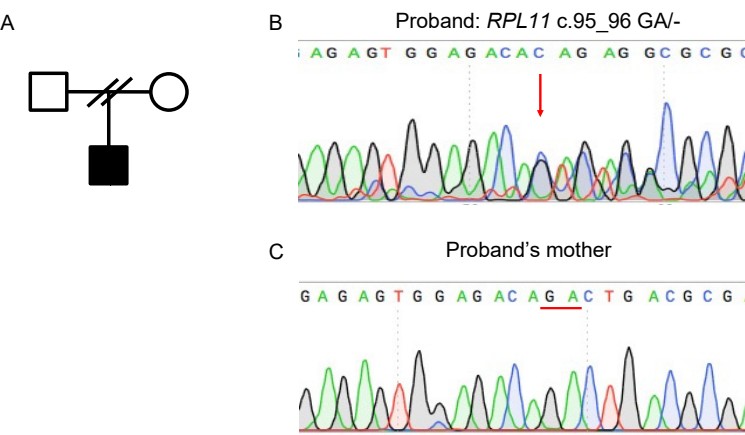

**Figure 1.** (**A**) Family pedigree; due to divorce of the proband's parents, no Sanger DNA sequencing was obtained from the proband's father. Sanger DNA sequencing of *RPL11* exon 2 from peripheral blood cells taken from the proband (**B**) and his mother (**C**). Sequences of both forward and reverse strands shown. Red arrow in (**B**) depicts where deletion began. Red line in (**C**) indicates reference alleles.

He is the sole child of a non-consanguineous marriage and a search for HLA-matched donors failed to identify any. However, we did find HLA-A/B/DR 5/6 (HLA-A/B/C/DQ/DR 7/10 by high resolution)-matched and ABO-mismatched (O to A) unrelated cord blood. At transplantation, he was 5.9 years old with a body surface area of 0.5 m$^2$ (body weight:

10.9 kg; height: 82.5 cm). For the transplant, we employed an individualized PK-guided conditioning regimen [9,10,12,14,15], as described in Section 2, which consisted of fludarabine/busulfan (with a total of 216/348 mg per m$^2$ divided into four daily doses from day-7) and thymoglobulin (with a total of 15 mg/kg divided into four daily doses from day-9). The dosage calculations of busulfan by TDM (Table 1) and of thymoglobulin/fludarabine by the respective population-based PK model are shown in Table 2. There was a 27.5% reduction in busulfan (compared to the empirical formulation of 480 mg/m$^2$ [17]) and a 35% increase in fludarabine (compared to the common formulation of 160 mg/m$^2$). The thymoglobulin dosage increased over three-fold (compared to the recommendation of 4.5 mg/kg) and was given from day-9 instead of from day-2. The estimated active thymoglobulin exposure for the patient is shown in Figure 2. For prophylaxis against graft-versus-host disease (GVHD), the patient received cyclosporine (from day -1) and methylprednisolone. Ursodiol was given to prevent sinusoidal obstruction syndrome. On 22 October 2018 (day 0), he received the cord blood unit containing $15.5 \times 10^7$/kg total nucleated cells and $4.5 \times 10^5$/kg CD4$^+$ cells. The neutrophil engrafted on day +13. He received a total of ten units red blood cells and three units of platelet transfusions peri-CBT at our hospital. No transfusions were required after day +15.

**Table 1.** Therapeutic drug monitoring (TDM) of busulfan. IV, intravenous; PK, pharmacokinetic.

| Time after IV Busulfan Infusion | TDM of Busulfan (ng/mL) | Prediction (ng/mL) | Fit to Population PK Model [12] |
|---|---|---|---|
| 5 min | 4297 | 3973.96 | Yes |
| 1 h | 2894 | 2949.63 | Yes |
| 2 h | 2061 | 2054.51 | Yes |
| 3 h | 1573 | 1544.6 | Yes |

**Table 2.** Dose estimation of conditioning regimens. AUC, area under curve; HCT, hematopoietic cell transplantation; PK, pharmacokinetic; TDM, therapeutic drug monitoring.

| Conditioning Regimen | Dose Determination Method | Recommended AUC | Actual AUC Used |
|---|---|---|---|
| Fludarabine | Estimated using a validated fludarabine PK model [8] according to body weight and serum creatinine. | 17.5–22.5 mg·/L | 17.5 mg·h/L |
| Busulfan | TDM, followed by validation with a Busulfan population PK model [12]. | 70–101 mg·h/L | 90 mg·h/L |
| Thymoglobulin | Estimated using a validated pediatric population PK model [13] according to body weight and pre-dose lymphocyte count. | >40 AU·day/mL pre-HCT; <20 AU·day/mL post-HCT | 30.7 AU·day/mL pre-HCT; 4.3 AU·day/mL post-HCT |

His treatment course was complicated by grade III acute GVHD (skin and gastrointestinal stage 3), with skin rashes appearing from day +19 and progressing into grade III at about day +41 post-transplant. This was resolved by increasing the steroid dosage (Figure 3). No sinusoidal obstruction syndrome, respiratory distress, or other organ dysfunctions were encountered. Cyclosporine was discontinued within 5 months post-CBT. However, he required prolonged maintenance steroids (corresponding to 12.5 mg twice daily from day +85) for tertiary adrenal insufficiency, which were successfully tapered off 13 months post-CBT. Thereafter, the patient displayed normal cortisol and adrenocorticotropic hormone levels. CD4$^+$ T cell reconstitution reached 58, 105, 1350 and 751 per μL on day +45, day +68, one-year, and more than 2.5 years post-CBT, respectively. No chronic GVHD, cytomegalovirus, or Epstein–Barr virus reactivations nor other serious infections were encountered.

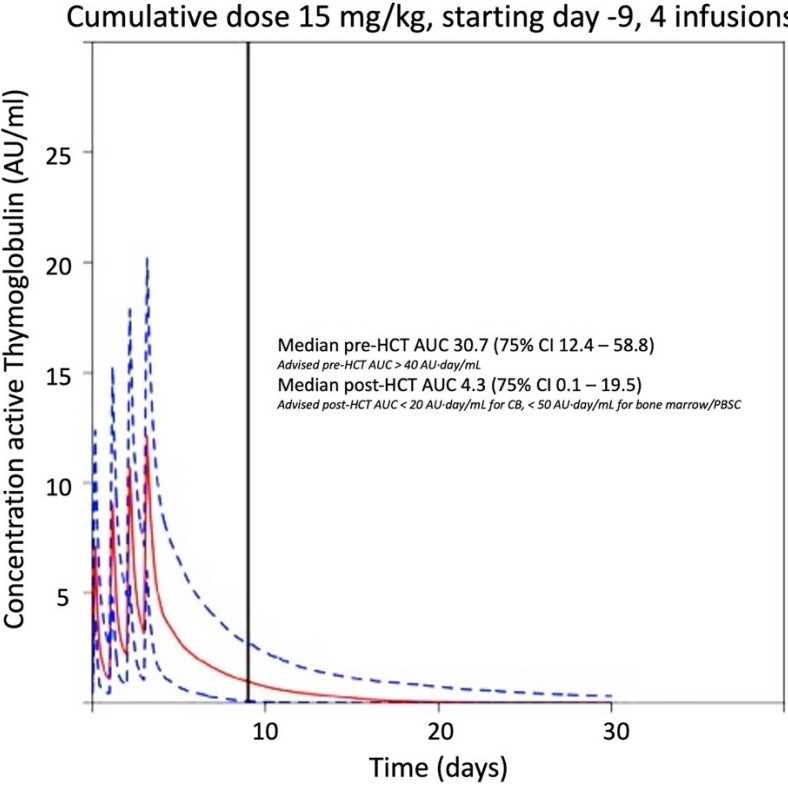

**Figure 2.** The estimated thymoglobulin exposure of the patient, targeting AUC of 30.7 AU·day/mL pre-HCT and of 4.3 AU·day/mL post-HCT, respectively, with a total of 15 mg/kg thymoglobulin divided into 4 daily infusions from day-9 to day-6. Black line, day 0 of HCT. Red line, median concentration of active thymoglobulin; blue dot lines, 75% confidence interval (CI). AUC, area under curve; CB, cord blood; HCT, hematopoietic cell transplantation; PBSC, peripheral blood stem cell.

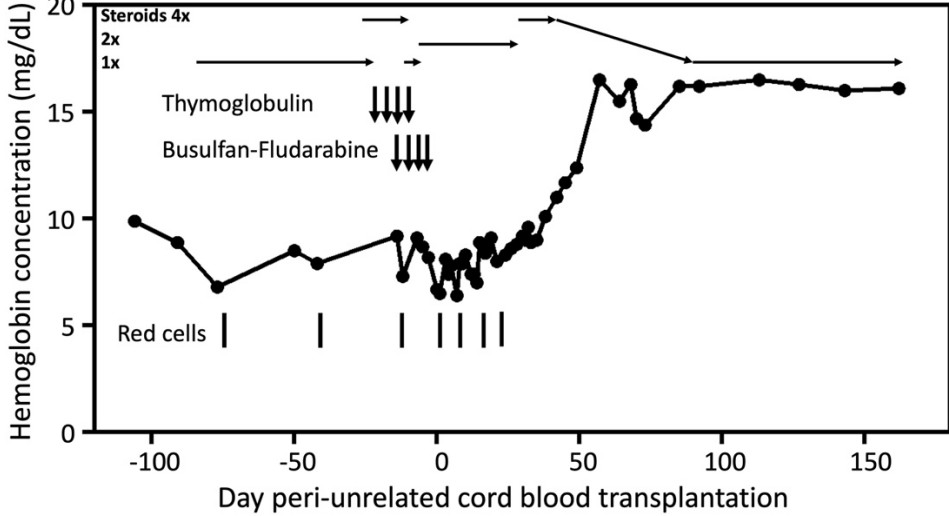

**Figure 3.** Peri-transplant hemoglobin concentrations and treatments of the patient, with day 0 indicating the day of cord blood infusion. Bars indicate red cells transfusions. Downward arrows indicate the days' conditioning agents (thymoglobulin, busulfan, and fludarabine) given. Horizontal arrows indicate the different comparative dosages of steroids given with 1× corresponding to maintenance 12.5 mg cortisone twice daily.

The patient has since returned to school. His height and weight have increased to 106.7 cm (from 80 cm) and 17.5 kg (from 11 kg), respectively, two years and 11 months post-CBT, but are still below the fifth percentile for age. He had 96–98% donor chimerism

and normal blood counts post-engraftment after transplant, with a white blood cell count of $8.98–10.16 \times 10^9$/L, a neutrophil count of $3.168–4.572 \times 10^9$/L, a hemoglobin concentration of 14.1–14.3 mg/dL, and platelets of $287–312 \times 10^9$/L during the year 2021. His ferritin level had elevated to 1388 ng/mL at four months post-CBT but then decreased to 337 ng/mL at 34 months post-CBT. His scoliosis has remained stable and he has been taking daily vitamin D and calcium supplements. Analyses of cardiorespiratory performance and other visceral and endocrine functions revealed outputs within normal ranges.

## 4. Discussion

This patient, a sporadic DBA case carrying an *RPL11* nonsense mutation, had suffered severe complications from early and prolonged steroid treatment. Although steroids were effective in terms of limiting transfusions, the treatment resulted in severe side effects, and it is now deemed an inappropriate approach by international consensus [18]. After switching to the transfusion program, the patient suffered a life-threatening adrenal crisis and a rapid evolution of iron overload, with the latter having been reported as more severe in DBA patients than in thalassemia or sickle cell anemia patients [19]. Thus, HCT was chosen to be the salvage option for the patient.

The results from HLA-matched HCT for DBA patients are satisfactory, with overall survival reaching over 90% [20–22]. Better outcomes have been achieved when the HCT was performed at a young age, and similar results have been attained with grafts from related versus unrelated donors as well as with myeloablative versus reduced-intensity conditioning regimens [20,22]. Similarly, HLA-matched sibling CBT has been highly successful for DBA patients [6,7].

However, related, or unrelated HLA-matched donors are often not available while unrelated CBT for DBA patients has proven challenging because of high rates of graft failure and other complications, such as sinusoidal obstruction syndrome and respiratory distress syndrome [6,7]. Nevertheless, successful outcomes from unrelated CBT after either reduced-toxicity or myeloablative conditioning for DBA have been reported [21,23,24]. For instance, two DBA patients carrying the *RPL5* and *RPS19* mutations, respectively, exhibited successful engraftment after reduced-intensity 5/6 HLA-antigen-matched unrelated CBT [22]. In addition, three DBA patients (including one documented carrying the *RPS19* mutation) were successfully treated with unrelated CBT by standard myeloablative conditioning [24].

To treat DBA with myeloablative unrelated CBT, it is likely important to optimize exposure to fludarabine, busulfan, and thymoglobulin as ribosomopathy could cause systemic disturbances affecting drug disposition. The associations of fludarabine overexposure with impaired immune reconstitution and underexposure with increased graft failure and transplantation-related mortality have been reported [9]. The optimum intravenous busulfan AUC of 78–101 mg·h/L has been demonstrated to result in a lower graft failure rate (compared to a low AUC group) and lower transplantation-related mortality (compared to a high AUC group) in children after HCT, including unrelated CBT [12]. In addition, the PK-based treatment model for thymoglobulin has shown that low thymoglobulin exposure after pediatric unrelated CBT was correlated with excellent immune reconstitution and survival [14,15]. There are marked differences between the optimizing dosages of three conditioning agents compared to those of standard myeloablative regimens that have been established to minimize the physiological degree of variability. This highly suggests that the drug disposition and pharmacokinetic effects are affected by DBA patient with a systemic ribosomopathy phenotype. Further works are needed to investigate how systemic ribosomopathy can affect the distribution and metabolism as well as the excretion of the conditioning agents. In this study, we adjusted the fludarabine dosage according to actual body weight and renal function, while the thymoglobulin dosage was determined according to body weight and pre-dose lymphocyte count. Although the concentrations of fludarabine and thymoglobulin were not monitored in the current patient, the dosages of both drugs based on the previously published population PK models, plus TDM of busulfan did help

the patient to present rapid engraftment, less toxicity, and adequate immune reconstitution, as well as an alleviation of steroid- and transfusion-associated complications.

**Author Contributions:** R.-L.C. and L.-Y.C. designed the study. R.-L.C., L.-H.F. and L.-Y.C. carried out the experiments and performed data analysis. R.-L.C. and L.-Y.C. wrote the manuscript. All authors have read and agreed to the published version of the manuscript.

**Funding:** This research was funded by the Family Association for Children with Serious Illness, Taiwan (Grant no.: FACSI 211001).

**Institutional Review Board Statement:** Approval code of Institutional Review Board of Koo Foundation Sun Yat-Sen Cancer Center: 20131209A.

**Informed Consent Statement:** The case consent form has been signed up in September 2018 and re-signed according to the new case consent form: 20211227A.

**Data Availability Statement:** Data is contained within the article.

**Acknowledgments:** We thank Arief Lalmohamed, Kim van der Elst, and Matthijs van Luin, pharmacists at the Department of Clinical Pharmacy, Division Laboratories, University Medical Centre Utrecht, Utrecht, The Netherlands, for detailed PK information on busulfan and fludarabine. Rick Admiraal (University Medical Centre Utrecht, Utrecht, The Netherlands) is kindly acknowledged for his assistance in thymoglobulin prediction dose. We thank Peng Peng Ip (Institute of Molecular Biology, Academia Sinica, Taipei, Taiwan) for data analysis.

**Conflicts of Interest:** The authors declare no conflict of interest.

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
