# Peer review of "Individualized Dosage Optimization for Myeloablative Conditioning before Unrelated Cord Blood Transplantation in a Diamond–Blackfan Anemia Patient with Germline RPL11 Mutation: A Case Study"

_processes, doi:10.3390/pr10020201_

Round 1
Reviewer 1 Report
This is a very interesting case report, which is relevant to the field of stem cell transplantation. It provides a nice example of the novel treatment improvement via targeting conditioning regimens. In this, previous research has shown that especially ATG, Fludarabine, and Busulfan should be targeted. The paper is well written and clear, and could be accepted in its present form. I have no comments to add.
Author Response
Reviewer 1
This is a very interesting case report, which is relevant to the field of stem cell transplantation. It provides a nice example of the novel treatment improvement via targeting conditioning regimens. In this, previous research has shown that especially ATG, Fludarabine, and Busulfan should be targeted. The paper is well written and clear, and could be accepted in its present form. I have no comments to add.
Response: Thank you.
Reviewer 2 Report
In my opinion this manuscript is completely beyond the scope of the journal. It should be sent to the journal concering the therapeutic drug monitoring, focusing on case reports.
Author Response
In my opinion this manuscript is completely beyond the scope of the journal. It should be sent to the journal concering the therapeutic drug monitoring, focusing on case reports.
Response: Thank you for reviewing our manuscript. Our aim is to publish this case study in the special issue “Studies of the dosage form and stability of the drug by various techniques” in the section “Pharmaceutical Processes” of the journal “Processes”. We try our best to illustrate the benefits of applying pharmaceutical optimization (drug dosage) instead of using standard conditioning regimens before unrelated cord blood transplantation for Diamond-Blackfan anemia patient in which poor outcome has been reported repeatedly. To make it clearer for readers, we have made major revisions.
Reviewer 3 Report
In this manuscript the authors present a case study of a patient suffering from Blackfan-Diamond anaemia that was treated with myeloablative cord blood transplantation. The dosage of conditioning drugs was individually optimized based on the previously published population pharmacokinetic (PK)models. As the authors argue, the dose optimization allowed for a safe and successful myeloablative cord blood transplantation.
Although the manuscript is a well-written and valuable example of a practical illustration how population PK modelling may translate to more personalized decision making in drug dosage, there are numerous points that raised my concern and, in my opinion, should be addressed by the authors before publication is recommended. The major points are listed below followed by some minor comments.
Major points:
- This manuscript is not a pharmacokinetic paper. It applies pharmacokinetic principles but the authors even do not confirm the that they obtained the parameter values they were aiming at. Only busulfan concentrations were measured but even this data is not at all to be found in the results section. The sections in Material and methods should be named e.g. “Fludarabine/Thymoglobulin dose optimization” rather than “PK”. The choice of the Journal is also puzzling for me as I could not find any keywords in Aims & scope that, in my opinion, would match this study (as said before, this is NOT a PK paper). This is a typical medical case study and this remark should be presented in the title not to mislead the readers.
- The highest value of this case lies in the presentation how population PK modelling can help to identify the true sources of variability and, based on that, to individualize the treatment in a given patient. Unfortunately, the authors were very frugal in words when it comes to HOW they actually arrived at these final dosage protocols. In my opinion this is the most important part of the paper and needs significant expansion. Based on the laconic information in the acknowledgements section I assume they had everything calculated by the colleagues from Utrecht and were only given the final value. Whereas from the clinical point of view this is completely fine, from the scientific point of view this is not satisfactory. If the authors do not feel competent in writing these paragraphs, I suggest inviting their colleagues from Utrecht to explain the PROCESS how they used the PK model, demographic data of the patient and thresholds in the PK parameters of interest in order to obtain the best dosage for the patient described in this case study.
- The limitations of this study should be clearly stated: the concentrations of two drugs were not confirmed, therefore, it is not known whether the parameters of interests were really achieved. We only know the successful clinical outcome but it’s not enough to claim that the dose optimization really worked as the authors intended.
Other points:
- The title should include the phrase “case study”
- Line (L) 15: inferior… to what? Please rephrase. Moreover, is that the DBA that directly affects the PK of the conditioning agents? Or is it a physiological degree of variability that would also be found in a normal population? How could DBA directly affect the disposition of the conditioning agents, please include this in your introduction or discussion.
- L25: “He” -> the patient
- L33: “severe anaemia of Diamond–Blackfan anemia” sounds awkward. Consider “severe haematological changes seen in the Diamond–Blackfan anaemia”.
- The introduction is too concise. Please introduce briefly the disease and the treatment options. In the end you suddenly mention the side effects of the GC therapy and blood transfusion. I think it would be better to first state what are the options in DBA (GC and transfusions) and when is the CBT recommended. This will explain the last sentence in the introduction which now is a bit surprizing for a reader not acquainted with the treatment of DBA and related diseases.
- L45-54: please provide more technical details (e.g. the equipment or software used).
- Points 2.2. and 2.4. Should be rather named “Dose optimization” as the current chapter titles are very misleading. Dosage means not only the dose but also the number of doses and the intervals. Please provide these details here. My other comments on providing more details on dose calculations are stated earlier.
- L56: typo in Utrecht
- L59: The reference says 20 mg*h/L. Why 17.5 then?
- L59: “the best”
- Point 2.3. provides a lot of technical details regarding drug analysis which is nice. However, the sample processing is not mentioned with a single word, why? Additionally, please provide the measurements of busulfan in the results section. How the monitoring allowed you to optimize the dosage?
- L76: missing units
- L89: The term AU (arbitrary units) is associated with the method of assessing the concentration of the drug. Without explanation it is not clear for the readers. Please explain how you arrived at this dose.
- Fig. 2 – Please explain what the black line is (you can put a label e.g. CBT or HCT on the graph, please keep the consistency in the nomenclature).
- The discussion is also quite concise. Please try to introduce the reader a bit more to the problem of the individualized therapy in haematological disorders. What are the sources of variability in drug PK? What should be included in the dose development to optimize the AUC? Is it eGFR? You never mention what was this value in the patient. Was it normal or abnormal?
- L207-209: the sentence lacks a verb
- L212: disposition (singular)
Round 2
Reviewer 2 Report
The explanations of Authors are clear. The manuscritpt is interesting. The alternations in the text improved its quality.
Reviewer 3 Report
Thank you for the corrections. The paper has been significantly improved.